# Effect of Temperature, Seed Size, Sowing Depth, and Position on Seed Germination and Seedling Growth of *Bauhinia retusa* Roxb. and *Bauhinia variegata* L.

**Neeraj Yadav [1], Vinod Prasad Khanduri [1], Bhupendra Singh [1,*], Chatar Singh Dhanai [1], Manoj Kumar Riyal [1], Deepa Rawat [1], Taufiq Ahmad [1] and Munesh Kumar [2,*]**

[1] Department of Forestry, College of Forestry, Veer Chandra Singh Garhwali Uttarakhand University of Horticulture and Forestry, Ranichauri, Tehri Garhwal 249199, Uttarakhand, India; neeraj.yadav52@gmail.com (N.Y.); khandurivp@yahoo.com (V.P.K.); dhanai_agro@rediffmail.com (C.S.D.); manojrayal1509@gmail.com (M.K.R.); rawatdeepa291@gmail.com (D.R.); ahmedmaliktaufiq@gmail.com (T.A.)

[2] Department of Forestry and Natural Resources, HNB Garhwal University, Srinagar 246174, Uttarakhand, India

\* Correspondence: butola_bs@yahoo.co.in (B.S.); muneshmzu@yahoo.com (M.K.)

**Abstract:** In urban forestry plantations are implemented in different cities of the world for social and environmental benefits. *Bauhinia retusa* and *Bauhinia variegata* are important species and to be used as large-scale plantation programs in urban forestry which might solve or mitigate urban, social, and environmental issues such as improving the physical & mental health of residents, food and nutrition security, increasing urban biodiversity, cooling the neighboring, preventing soil erosion, flooding, and mitigating greenhouse gas emissions and air pollution. The present study was conducted with the aim of producing quality planting material for *B. retusa* and *B. variegata* in the nursery for afforestation programs. Seeds of *B. retusa* and *B. variegata* were collected from the natural habitats to assess seed germination and seedling growth. Seeds were stored in different types of containers at room temperature and later on exposed to 15, 20, and 25 °C in seed germinator. Seeds were further sown in polythene bags according to the seed size, seed coat color, seed sowing depth, the orientation of seeds, and the result of the emergence of seedlings, their growth, and biomass were estimated. A two-way analysis of variance was calculated to estimate the variation among the studied parameters. Results revealed that a constant 25 °C temperature was considered best for seed germination of both the *Bauhinia* species. Polybags were found the most suitable for storing the *Bauhinia* seeds among the storage containers. The seedling emergence and growth were maximum in yellow color and large seeds. In *B. retusa*, seedling emergence, and growths were the maximum in seeds sown at a horizontal position and in *B. variegata* at an upright position. Seedling emergence, length, and biomass were recorded the maximum when seeds of *B. retusa* were sown at 4 cm depth and *B. variegata* seeds were sown at 2 cm depth. The study recommends that the yellow color seed that has to be sown at 2 cm to 4 cm depth with upright and horizontal positions is considered best for the production of quality planting stock of both studied *Bauhinia* species.

**Keywords:** forest trees; sub-tropical species; germination; seed size; seed coat color; sowing depth; sowing orientation

## 1. Introduction

*Bauhinia* belongs to the family Fabaceae and is a large genus of flowering plants that naturally occur in subtropical forest ecosystems with various forms such as trees, shrubs, and climbers [1]. Fifteen species of *Bauhinia* are generally grown in India [2]. *Bauhinia retusa* Roxb. and *Bauhinia variegata* L. are ecologically and economically important species. Farmers of the Himalayan region grow both species in different agroforestry systems [3],

and harvested for good quality fodder during the lean period, particularly in the summer season [4]. Farmers harvest the foliage of these trees to feed their cows, ox, and buffaloes. The local inhabitants of the Himalayan region collect *B. variegata* flower buds for making vegetables and pickles. Consequently, seed formation is declining and reducing the natural regeneration of the species [5]. The pods of both *Bauhinia* species are dehiscent, the seeds are flat and the epigeal type of germination is has occurred [5]. The new seedling in nature is hindered due to the over-exploitation of fodder and firewood. *B. variegata* contains many secondary metabolites in their flowers, flower buds, stems, roots, bark, seeds, and leaves and is used as a medicine for the prevention of various diseases [6,7]. Both *Bauhinia* species are most suitable for avenue plantation in urban areas. They produce beautiful flowers and leaves which are prerequisites for aesthetic plantation. Due to nitrogen fixation ability, both tree species are also good for the improvement of the soil quality and rehabilitation of land in urban areas [4]. The planting of trees in urban forestry programs provides different direct and indirect benefits. The direct benefits are in terms of economic as well as social well-being and indirect benefits as the environmental conditions of the region are modified. Thus, afforestation or plantation programs of any species, quality planting material is required. The quality of planting material grown in the nursery depends on the seed quality, time of sowing, and methods of sowing [8].

The emergence of the seedling is a remarkable process that is resolved by proper timing, location of emergence, and burial depth in the soil [5]. The seeds buried at different soil depths may experience unusual environmental situations such as oxygen, carbon dioxide, temperature, water, and nutrition deficiency affecting seed emergence and seedling growth [9–11]. Seed germination depends on several ecological conditions, i.e., light, temperature, and moisture. The color of the seed coat is a trait associated with water assimilation in seeds [12,13]. The water imbibition rate of several legume species depends on the seed coat color [12]. The seed coat development is related to the kind of genotypes, but environmental conditions such as photoperiod influence the color of the seed coat [12]. Bewley and Black [14] reported that the green seeds of *Salsola volkensii* Asch. & Schweinf. (Amaranthaceae) did not have dormancy, while the non-green seeds showed dormancy. The orientation of seed sowing, especially in large-seeded species, influences seedling emergence both on the forest floor and in the nursery. Prasad and Nautiyal [15] reported that the seedling emergence in *B. retusa* is affected by seed orientation in the nursery seed bed. Planting seeds of these species in nurseries recorded variable germination, while low emergence from the forest floor. Seed size is a heritable trait and is used to improve the performance of seeds for the future. A significant relationship between seed size and weight on seed germination and seedling growth in different forest tree species was observed [16–18]. Seed sowing depth plays a significant role in germination and seedling growth [19,20]. The seeds buried at shallower depths have the chance to be taken by predators. Seeds buried deeply spend extra reserves of food materials in elongation of stem for their shoot apices to emerge at the soil surface than seeds sown at shallower depths. However, seeds disclosed to air on the soil surface or forest litter are disposed of and lose viability due to rapid moisture loss [9]. Thus, the maximum emergence of a seedling is dependent on the sowing depth and orientation of the seeds. Seed sowing in shallow or too deep with an incorrect position would delay the emergence and growth of seedlings. An assessment of the relative impact of temperature, seed size, and sowing position on seed germination and seedling growth of *B. retusa* and *B. variegata* is of high significance when focussing on the production of good quality seedlings in short durations in the nursery. Therefore, the present investigation was conducted to find the answer to the following questions (i) Does the temperature, seed size, color of the seed coat, and period of storage influence the seed germination of *B. retusa* and *B. variegata* ? and (ii) Do the seed orientation (positioning) seed depth of sowing affects the seedling emergence, growth, and biomass accumulation of both *Bauhinia* species?

## 2. Materials and Methods

Mature pods of *B. retusa* and *B. variegata,* which were hard, flat, and brown in appearance, were collected from the trees by direct climbing from the Nagni area of Tehri district located, between 30° 16′15″ N latitude and 78°22′01″ E longitude, at an altitude of 890 m a.s.l. After seed collection, all pods were kept in sunlight for dehiscence till the seeds separated from the pod. Healthy seeds were separated and stored at room temperature (average 22 °C) for further study.

Thirty seeds (three replicates of ten seeds each) of both *Bauhinia* species were dipped in distilled water for 24 h before placing them into Petri plates (12 cmdiameter.) containing Whatman No.1 filter paper (two layers) and placed in pre-fixed temperatures. The temperatures used were 15, 20, and 25 °C in the seed germinator under a completely randomized design. The humidity in the germinator was pre-fixed at 70.0%. The emergence of the radicle was considered an indicator of the germination of seeds. The period for germination was 28 days, and five seedlings were selected from each replicate to measure the radicle and plumule length with the help of a meter scale in centimeters (cm). Mean Germination Time (MGT) was calculated as per the formula given by Ellis and Roberts [21].

$$MGT = \frac{\sum Dn}{\sum n}$$

Germination Index (GI) was estimated as per the formula given by Kendrick and Frankland [22].

$$\text{Germination index} = \frac{Total\ percent\ germination}{Time\ (h)taken\ for\ 50\%\ germination}$$

Three different types of storage containers, i.e., polybags, canes, and cotton bags, were used for storing the seeds of both *Bauhinia* species. Four hundred seeds were retained in each type of container at room temperature for one year in the Laboratory, College of Forestry, Ranichauri, Tehri Garhwal, Uttarakhand, to evaluate the storage behavior of *Bauhinia* seeds.

Ninety seeds were placed in three replicates (30 seeds each) in Petri dishes lined with Whatman No. 1 filter paper, at 25 °C, after 3, 6, 9, and 12 months of storage in each type of container. When the radicle had emerged, seeds were considered to germinate. Germination percentage, MGT, GI, radicle, and plumule length were calculated as per the above-given methods.

Grading of seeds was executed based on their size and seed coat color to assess the effect of the seed size and seed coat color on germination, survival, and growth of seedlings. Seeds of both species were sown in the nursery in pre-filled poly bags with a mixture of a 1:2:1 ratio of sand, soil, and Farm Yard Manure (FYM). A hundred seeds of each species, five replicates (20 seeds each) were sown in the poly bags according to seed size (large and small), seed coat color (brown and yellow), seed sowing depth (2, 4, and 6 cm), and seed orientation (micropylar up, micropylar down and horizontal radicle end at 90° in respect to soil surface). The emergence of seedlings was recorded every day for 30 days or one week further if germination continued. Weeding and watering in the poly bags were executed manually when desirable till 12 months. Twenty seedlings of each treatment, in each of the five replications, were randomly tagged to assess 12 months growth, and morphological traits *viz* shoot length (cm), collar diameter (mm), number of leaves per seedling, root length (cm), shoot dry weight (g), root dry weight (g), leaves dry weight (g), and total dry weight (g) measured as per the standard procedures.

Statistical analysis: Standard derivation (±SD) and two-way analysis of variance (ANOVA) were calculated to analyze the effects of different seed sizes, seed coat color, and sowing treatments on germination, shoot length, root length, shoot dry weight, root dry weight, leaves dry weight and total dry weight of both *Bauhinia* species by using the statistical software WASP version 1.0 (online software package), ICAR, GOA, India, and

the level of significant were tested at $p < 0.01$ and $p < 0.05$. Duncan's multiple range tests were used to determine significant differences between different treatments at a 5% level.

## 3. Results

Seeds of two *Bauhinia* species were tested for germination at constant temperatures were 15, 20, and 25 °C. Significant variation ($p < 0.05$) was found in seed germination at different constant temperatures. The radicle and plumule growth were also significantly ($p < 0.05$) different at different constant temperature regimes. Maximum germination, radicle, and plumule length were recorded at 25 °C, and the lowest germination percent, radicle, and plumule length were recorded at 15° C in both species (Table 1).

**Table 1.** Effects of temperatures on germination, radicle, and plumule growth (Mean ± SD) of two *Bauhinia* species (Mean values followed by the same letter within the column are not significant ($p < 0.05$) among the temperature treatments).

| Species | Temp. (°C) | Germination (%) | Radicle (cm) | Plumule (cm) |
|---|---|---|---|---|
| *Bauhinia retusa* | 25 °C | 83.33 ± 5.77 [a] | 8.98 ± 0.57 [a] | 3.28 ± 0.20 [a] |
| | 20 °C | 66.67 ± 11.55 [b] | 6.74 ± 0.96 [ab] | 2.81 ± 0.62 [ab] |
| | 15 °C | 60 ± 10.0 [b] | 5.89 ± 0.68 [b] | 1.36 ± 0.22 [b] |
| *Bauhinia variegata* | 25 °C | 86.67 ± 5.77 [a] | 9.40 ± 0.52 [a] | 3.52 ± 0.11 [a] |
| | 20 °C | 73.33 ± 5.77 [ab] | 6.94 ± 0.62 [b] | 2.81 ± 0.62 [ab] |
| | 15 °C | 70.0 ± 10.0 [b] | 5.81 ± 0.29 [b] | 1.49 ± 0.13 [b] |

Significant variation ($p < 0.05$) was found in seed germination after the seed was stored in different types of containers. After one year of storage of *B. retusa* seeds, a maximum of 50.0% germination was recorded in seeds stored in a cotton bag, and a minimum of 43.0% was in polybags. While *B. variegata* seeds showed the highest longevity in terms of germination in polybags and the lowest in cotton bags (Table 2). The radicle and plumule growth of both species were also recorded and found that plumule and radicle growth were significantly ($p < 0.05$) reduced after increasing the period of storage. The type of containers also significantly ($p < 0.05$) influenced the radicle and plumule growth of *B. retusa* (Table 3).

**Table 2.** Germination percentage (Mean ± SD) of *Bauhinia retusa* and *B. variegata* seeds stored on different types of containers (Mean values followed by the same letter within the row are not significant ($p < 0.05$) among the types of containers).

| Species | Containers | | | |
|---|---|---|---|---|
| | Months | Cotton Bag | Plastic Cane | Polybags |
| *Bauhinia retusa* | 3 | 76.67 ± 11.56 [a] | 86.57 ± 5.77 [b] | 86.67 ± 5.77 [b] |
| | 6 | 56.67 ± 5.77 [b] | 66.67 ± 5.77 [ab] | 70.0 ± 10.0 [a] |
| | 9 | 50.0 ± 10.0 [b] | 53.33 ± 5.77 [ab] | 66.67 ± 5.77 [a] |
| | 12 | 48.0 ± 6.58 [b] | 50.0 ± 10.0 [ab] | 56.67 ± 11.55 [a] |
| *Bauhinia variegata* | 3 | 80.0 ± 10.0 [b] | 86.67 ± 5.77 [a] | 83.33 ± 5.77 [ab] |
| | 6 | 76.67 ± 5.77 [b] | 80.0 ± 10.00 [ab] | 83.33 ± 11.55 [a] |
| | 9 | 66.67 ± 5.77 [b] | 73.33 ± 5.77 [a] | 80.0 ± 10.0 [a] |
| | 12 | 60.00 ± 10.00 [b] | 65.00 ± 10.0 [a] | 75.0 ± 11.56 [a] |

Seed germinations were significantly ($p < 0.05$) influenced by different seed sizes, colors, and seed-sowing treatments in *B. retusa*. Large-size seeds have a higher germination

percentage than small-size seeds in both species (Figure 1). The value of MGT was 14.5 days for large-size seeds and 18.3 days for small-size seeds (Figure 2). GI values were 0.33 and 0.28 for large and small-size seeds, respectively (Figure 3). Similarly to the results of *B. retusa,* the value of MGT was the maximum (22.66 days) for small-size seeds and the minimum (18.23 days) for large-size seeds in *B. variegata* (Figure 2). The maximum and minimum (0.23 and 0.21) GI values were found for large and small-size seeds, respectively (Figure 3). On average, yellow color seeds produced a maximum of 94.4% and 78.4% germination in *B. retusa and B. variegata*, respectively. While the minimum of 79.6% and 66.8% germination were recorded in brown color seed coats, respectively in both species (Figure 1). MGT and GI were recorded as the maximum in yellow color seeds as compared to brown color seeds (Figures 2 and 3).

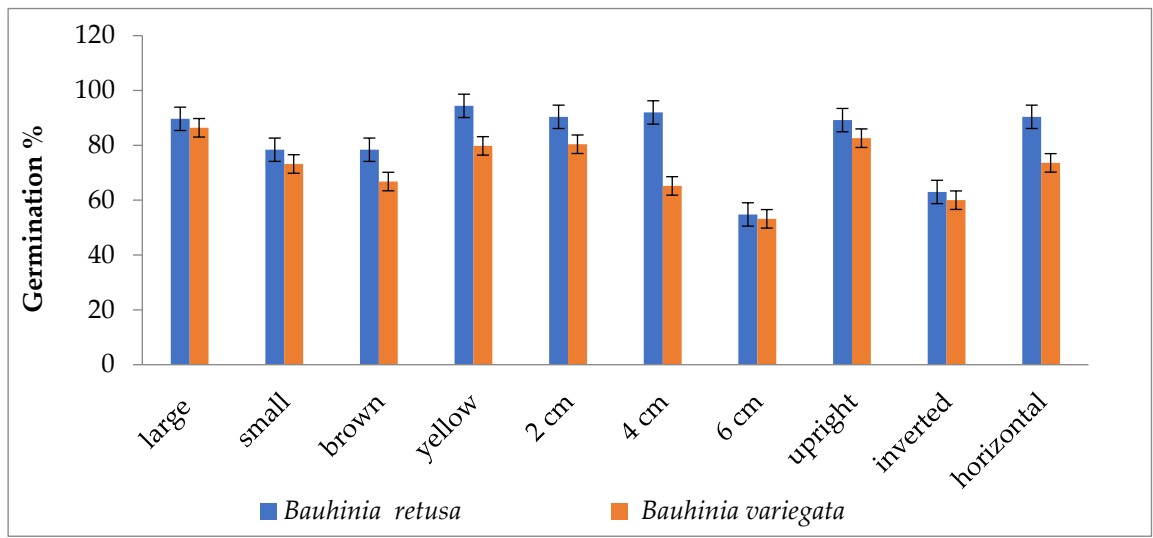

**Figure 1.** Germination percentage of *Bauhinia retusa* and *B. variegata in* different seed sizes, seed coat color, sowing depth, and sowing orientation (vertical bars representing ± S.D).

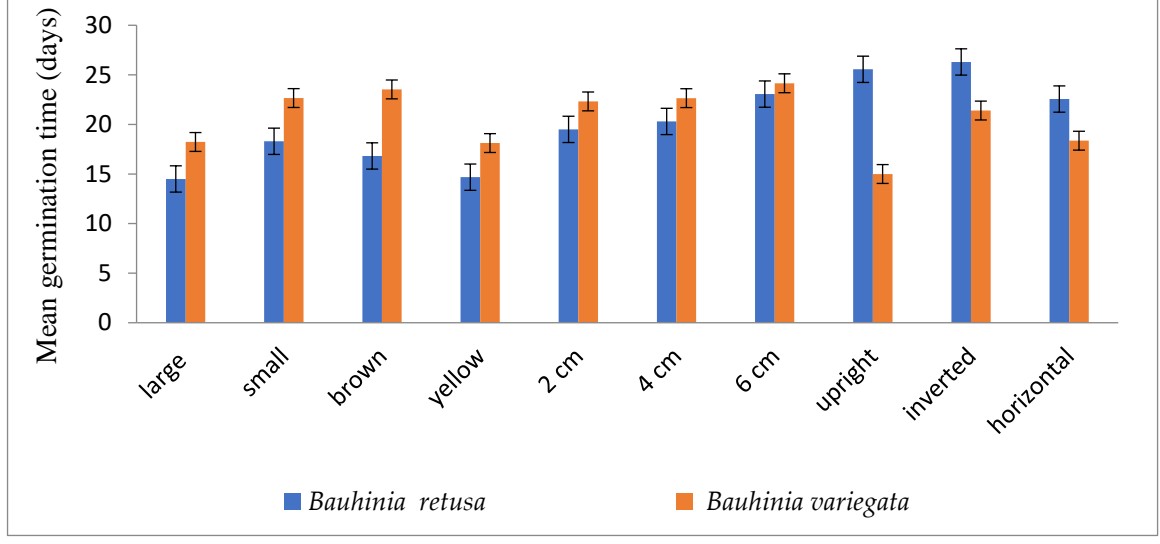

**Figure 2.** Mean germination time of *Bauhinia retusa* and *B. variegata* in different seed sizes, seed coat color, sowing depth, and sowing orientation (vertical bars representing ± S.D).

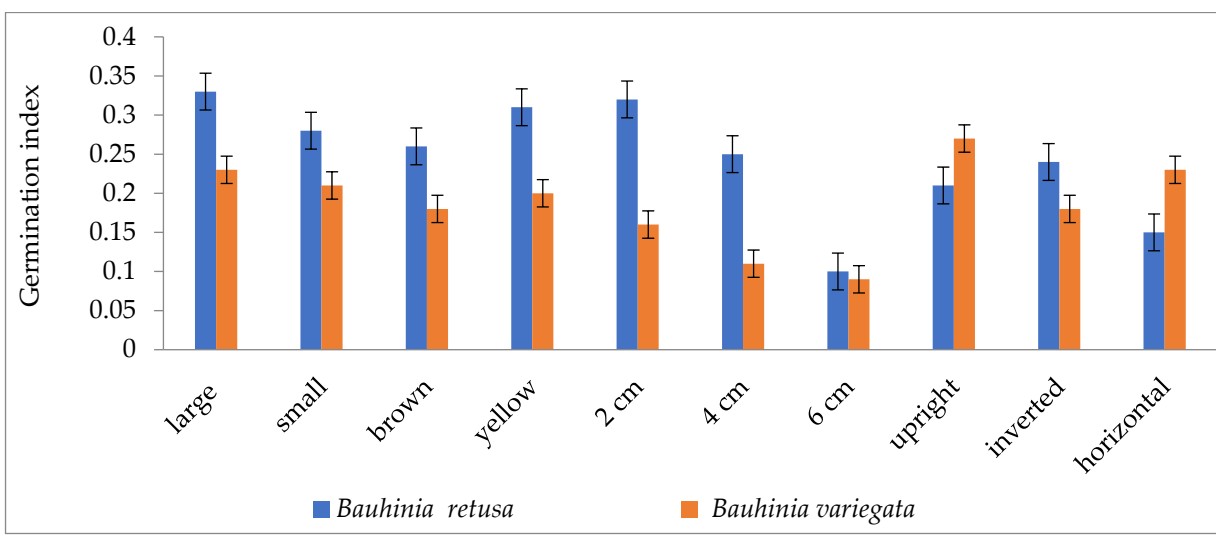

**Figure 3.** Germination index of *Bauhinia retusa* and *B. variegata* in different seed sizes, seed coat color depth, and sowing orientation (vertical bars representing ± S.D).

Seeds that were sown at 4 cm depth recorded a maximum of 92.0 and 82.4% emergence of seedlings and a minimum of 54.8 and 60.0% at 6 cm depth, respectively, for *B. retusa* and *B. variegata* (Figure 1). The MGT was the highest (23.0 and 24.15 days) at 6 cm depth, and the lowest (19.5 and 22.32 days) at 2 cm depth for both species (Figure 2). GI values were the highest (0.32 and 0.27) at 2 cm depth and the lowest (0.10 and 0.18) at 6 cm depth (Figure 3).

The seeds of *B. retusa* sown in different orientations, upright, inverted, and horizontal revealed that the horizontal sowing orientation had the highest (90.40%) germination and the lowest (63.0%) observed at inverted seed sowing orientation (Figure 1). The MGT was observed as the highest (21.4 days) in an inverted position, and the lowest (15.0 days) in an upright position (Figure 2). GI values were highest (0.20) in an inverted position and lowest (0.16) in a horizontal position (Figure 3).

Shoot length, collar diameter, number of leaves per plant, and root length were recorded as maximum (20.24 cm, 3.31 mm 2.60, and 29.56 cm, respectively) in the seedlings produced by large size seeds, and the minimum (17.77 cm, 2.75 mm, 2.28 and 15.75 cm, respectively) in small size seeds, after 12 months of growth of *B. retusa*. The large seeds also produced higher biomass as compared to the smaller size seeds. The dry weight of shoot, root, leaves, total dry weight, and root/shoot dry weight ratio wasnoted as the maximum (0.54 g, 1.78 g, 0.50 g, 2.83 g, and 3.29 g, respectively) in large, and the minimum (0.43 g, 1.36 g, 0.47 g, 2.261 g, and 3.16 g) in small size seeds (Table 4).

Length of the shoot, collar diameter, number of leaves per plant, root length, and dry weight of shoot, root, and leaves in *B. retusa* was the maximum at the upright position, and the minimum observed in the inverted position (Table 4). However, in *B. variegata* seedling growth and biomass production were recorded as the maximum in the horizontal position, and the minimum in the inverted position (Table 5).

**Table 3.** Radicle and plumule growth (Mean ± SD) of *Bauhinia retusa* and *B. variegata* seeds stored in different types of containers (Mean values followed by the same letter within the column are not significant ($p < 0.05$) among the types of containers.

| Containers | *B. retusa* | | | | | | | | *B. variegata* | | | | | | | |
|---|---|---|---|---|---|---|---|---|---|---|---|---|---|---|---|---|
| | 3 Month | | 6 Month | | 9 Month | | 12 Month | | 3 Month | | 6 Month | | 9 Month | | 12 Month | |
| | Radicle (cm) | Plumule (cm) | Radicle (cm) | Plumule (cm) | Radicle (cm) | Plumule (cm) | Radicle (cm) | Plumule (cm) | Radicle (cm) | Plumule (cm) | Radicle (cm) | Plumule (cm) | Radicle (cm) | Plumule (cm) | Radicle (cm) | Plumule (cm) |
| Polythene Bag | 11.97 [a] ± 1.22 | 4.11 [a] ± 0.50 | 8.52 [a] ± 1.51 | 3.11 [a] ± 0.11 | 6.94 [b] ± 0.52 | 1.57 [b] ± 0.32 | 5.64 [ab] ± 0.42 | 1.45 [b] ± 0.61 | 12.60 [ab] ± 1.11 | 5.16 [ab] ± 0.29 | 12.59 [a] ± 1.10 | 5.62 [a] ± 0.72 | 10.70 [a] ± 0.75 | 5.45 [a] ± 0.55 | 9.78 [a] ± 0.20 | 5.12 [a] ± 0.31 |
| Plastic cane | 9.77 [b] ± 0.48 | 3.50 [ab] ± 0.32 | 6.88 [b] ± 0.84 | 2.47 [b] ± 0.57 | 5.13 [b] ± 0.83 | 2.03 [a] ± 0.37 | 5.11 [b] ± 0.25 | 1.84 [a] ± 0.64 | 10.14 [b] ± 0.30 | 4.75 [b] ± 0.64 | 10.86 [b] ± 0.98 | 5.50 [b] ± 0.59 | 9.93 [b] ± 0.50 | 4.52 [b] ± 0.28 | 8.84 [b] ± 0.27 | 4.24 [b] ± 0.54 |
| Cotton Bag | 10.33 [ab] ± 0.87 | 3.26 [b] ± 0.26 | 7.45 [ab] ± 0.99 | 2.55 [b] ± 0.10 | 7.14 [a] ± 0.22 | 1.70 [ab] ± 0.50 | 6.84 [a] ± 0.78 | 1.67 [ab] ± 0.27 | 13.52 [a] ± 1.0 | 5.21 [a] ± 0.39 | 11.74 [ab] ± 0.82 | 5.52 [ab] ± 0.38 | 10.01 [b] ± 0.49 | 5.01 [ab] ± 0.20 | 9.14 [ab] ± 0.42 | 4.94 [ab] ± 0.18 |

**Table 4.** Growth attributes and their biomass in different sizes, sowing orientation, seed color, and sowing depth after 12 months of growth in *B. retusa* (Mean values followed by the same letter within the column are not significant ($p < 0.05$) among the sowing conditions/treatments).

| | 12 Months | | | | | | | | |
|---|---|---|---|---|---|---|---|---|---|
| Sowing Conditions | Shoot Length (cm) | Collar Diameter (mm) | Number of Leaves | Root Length (cm) | Shoot Dry Weight (g) | Root Dry Weight (g) | Leaf Dry Weight (g) | Total Dry Weight (g) | Root/Shoot Dry Ratio |
| | *Seed size* | | | | | | | | |
| Large | 20.24 [a] ± 1.53 | 3.31 [a] ± 0.21 | 2.60 [a] ± 0.14 | 29.56 [a] ± 2.18 | 0.54 [a] ± 0.07 | 1.78 [a] ± 0.21 | 0.50 [a] ± 0.08 | 2.83 [a] ± 0.36 | 3.29 [a] |
| Small | 17.77 [a] ± 1.73 | 2.75 [a] ± 0.66 | 2.28 [b] ± 0.18 | 15.75 [b] ± 3.89 | 0.43 [b] ± 0.11 | 1.36 [b] ± 0.11 | 0.47 [a] ± 0.08 | 2.26 [b] ± 0.31 | 3.16 [b] |
| | *Seed Orientation* | | | | | | | | |
| Upright | 20.68 [a] ± 2.93 | 3.29 [a] ± 0.09 | 2.40 [a] ± 0.14 | 29.61 [ab] ± 3.96 | 0.56 [a] ± 0.07 | 1.79 [a] ± 0.20 | 0.54 [a] ± 0.09 | 2.90 [a] ± 0.37 | 3.19 [b] |
| Inverted | 17.27 [b] ± 1.26 | 3.08 [b] ± 0.26 | 2.28 [b] ± 0.11 | 21.04 [b] ± 4.27 | 0.40 [b] ± 0.12 | 1.34 [b] ± 0.09 | 0.42 [b] ± 0.08 | 2.17 [b] ± 0.29 | 3.35 [a] |
| Horizontal | 18.74 [ab] ± 1.10 | 3.11 [ab] ± 0.27 | 2.32 [ab] ± 0.18 | 30.36 [a] ± 2.53 | 0.48 [ab] ± 0.12 | 1.57 [ab] ± 0.21 | 0.53 [ab] ± 0.10 | 2.58 [ab] ± 0.43 | 3.27 [ab] |
| | *Seed Colour* | | | | | | | | |
| Brown | 15.91 [b] ± 1.63 | 2.60 [a] ± 0.41 | 2.28 [a] ± 0.11 | 14.48 [b] ± 2.10 | 0.43 [b] ± 0.10 | 1.36 [b] ± 0.11 | 0.46 [a] ± 0.08 | 2.25 [b] ± 0.29 | 3.0 [b] |
| Yellow | 20.13 [a] ± 2.79 | 2.98 [a] ± 0.56 | 2.20 [b] ± 0.24 | 30.59 [a] ± 2.35 | 0.57 [a] ± 0.06 | 1.71 [a] ± 0.30 | 0.50 [a] ± 0.07 | 2.78 [a] ± 0.43 | 3.16 [a] |
| | *Sowing Depth* | | | | | | | | |
| 2 cm | 17.38 [a] ± 1.11 | 2.87 [ab] ± 0.35 | 2.44 [ab] ± 0.40 | 30.10 [a] ± 1.39 | 0.59 [a] ± 0.04 | 1.65 [a] ± 0.24 | 0.53 [a] ± 0.10 | 2.74 [a] ± 0.43 | 2.77 [b] |
| 4 cm | 13.63 [ab] ± 1.91 | 2.98 [a] ± 0.14 | 2.55 [a] ± 0.32 | 29.09 [ab] ± 2.48 | 0.49 [ab] ± 0.09 | 1.64 [ab] ± 0.33 | 0.51 [ab] ± 0.07 | 2.68 [ab] ± 0.43 | 3.36 [a] |
| 6 cm | 9.44 [b] ± 0.60 | 2.39 [b] ± 0.03 | 2.32 [b] ± 0.23 | 21.80 [b] ± 0.98 | 0.44 [b] ± 0.11 | 1.39 [b] ± 0.09 | 0.45 [b] ± 0.09 | 2.28 [b] ± 0.30 | 3.15 [ab] |

Values are Mean ± SD.

**Table 5.** Growth attributes and their biomass in different sizes, sowing orientation, seed color, and sowing depth, after 12 months of growth in *B. variegata* (Mean values followed by the same letter within the column are not significant ($p < 0.05$) among the sowing conditions /treatments).

| Sowing Condition | Shoot Length (cm) | Collar Diameter (mm) | Number of Leaves | Root Length (cm) | Shoot Dry Weight (g) | Root Dry Weight (g) | Leaf Dry Weight(g) | Total Dry Weight(g) | Root/Shoo Dry Ratio |
|---|---|---|---|---|---|---|---|---|---|
| | | | | **12 Months** | | | | | |
| | | | | Seed size | | | | | |
| Large | 32.09 [a] ±1.72 | 3.71 [a] ± 0.19 | 13.43 [a] ± 0.43 | 31.46 [a] ± 2.26 | 0.44 [a] ± 0.08 | 0.30 [a] ± 0.06 | 0.30 ± 0.04 [a] | 1.04 [a] ± 0.16 | 0.68 [b] |
| Small | 13.21 [b] ±1.94 | 2.64 [b] ± 0.35 | 5.17 [b] ± 0.67 | 24.54 [b] ± 4.67 | 0.24 ± 0.06 [b] | 0.19 ± 0.05 [b] | 0.20 ± 0.04 [b] | 0.60 [b] ± 0.11 | 0.79 [a] |
| | | | | Seed Orientation | | | | | |
| Upright | 34.52 [ab] ± 4.27 | 3.16 ± 0.21 [ab] | 8.10 ± 1.17 [a] | 30.68 ± 3.73 [a] | 0.47 ± 0.08 [ab] | 0.33 ± 0.06 [b] | 0.30 ± 0.04 [b] | 1.10 ± 0.15 [b] | 0.70 [b] |
| Inverted | 21.53 [b] ± 1.25 | 2.98 ± 0.16 [b] | 7.60 ± 0.67 [b] | 24.07 ± 3.07 [b] | 0.42 ± 0.08 [b] | 0.31 ± 0.05 [b] | 0.30 ± 0.04 [b] | 1.04 ± 0.13 [b] | 0.73 [a] |
| Horizontal | 41.92 [a] ± 2.94 | 3.43 ± 0.15 [a] | 8.18 ± 0.69 [a] | 31.63 ± 2.78 [a] | 0.51 ± 0.08 [a] | 0.37 ± 0.06 [a] | 0.35 ± 0.04 [a] | 1.23 ± 0.16 [a] | 0.72 [a] |
| | | | | Seed Colour | | | | | |
| Brown | 36.17 [b] ± 1.48 | 3.21 [a] ± 0.20 | 9.32 [a] ± 1.52 | 24.86 [b] ± 5.01 | 0.45 [b] ± 0.06 | 0.30 [b] ± 0.05 | 0.29 [b] ± 0.03 | 1.04 [b] ± 0.11 | 0.66 [b] |
| yellow | 40.07 [a] ± 1.70 | 3.58 [a] ± 0.21 | 10.22 [a] ± 0.63 | 33.72 [a] ± 3.09 | 0.50 [a] ± 0.07 | 0.36 [a] ± 0.06 | 0.33 [a] ± 0.04 | 1.20 [a] ± 0.12 | 0.72 [a] |
| | | | | Sowing Depth | | | | | |
| 2 cm | 31.70 [a] ± 4.71 | 3.33 [ab] ± 0.23 | 7.68 [a] ± 0.62 | 31.36 [a] ± 1.92 | 0.38 [a] ± 0.06 | 0.30 [a] ± 0.06 | 0.29 [a] ± 0.05 | 0.97 [a] ± 0.10 | 0.78 [b] |
| 4 cm | 18.48 [b] ± 1.99 | 3.37 [a] ± 0.15 | 7.66 [ab] ± 0.87 | 29.47 [ab] ± 1.71 | 0.35 [ab] ± 0.06 | 0.28 [ab] ± 0.04 | 0.27 [ab] ± 0.06 | 0.89 [ab] ± 0.09 | 0.80 [ab] |
| 6 cm | 17.62 [b] ± 2.54 | 3.28 [b] ± 0.12 | 7.48 [b] ± 0.66 | 20.39 [b] ± 3.38 | 0.27 [b] ± 0.08 | 0.23 [b] ± 0.07 | 0.26 [b] ± 0.05 | 0.75 [b] ± 0.15 | 0.85 [a] |

Values are Mean ± SD.

The yellow color seeds had gained the maximum growth attributes and biomass allocations as compared to the brown color seeds in both *Bauhinia* species. Shoot length, collar diameter, root length, number of leaves per plant, and biomass production were recorded as the maximum at 2 cm depth in both species, except collar diameter and number of leaves per plant, which were found at 4 cm depth in *B. retusa*. Large seeds produced the maximum growth attributes and biomass after 12 months as compared to the small size seeds (Tables 4 and 5).

The ANOVA was significantly ($p < 0.05$, 0.01) different among various treatments. Collar diameter was significantly ($p < 0.01$) affected by seed size, seed coat color, and sowing orientation treatments, except sowing depth in *B. variegata*, while in *B. retusa* sowing depth significantly affected the collar diameter. Seed sizes significantly ($p < 0.01$) enhanced the number of leaves per plant in *B. variegata*. However, the number of leaves was significantly influenced by the seed size and sowing depth in *B. retusa* (Tables 6 and 7).

**Table 6.** Two-way analysis of variance (ANOVA) for seed germination, growth attributes, and biomass allocation of *B. retusa* under different seed sizes, color, sowing depths, and sowing orientations.

| Source of Variation | DF | Germination (%) | Shoot Length (cm) | Collar Diameter (mm) | Number of Leaves | Root Length (cm) | Shoot Dry Wt (g) | Root Dry Wt (g) | Leaves Dry Wt (g) | Total Dry Wt (g) |
|---|---|---|---|---|---|---|---|---|---|---|
| Seed size | 1 | 3.13.60 ** | 15.25 NS | 0.80 NS | 0.25 * | 476.93 ** | 0.23 * | 0.44 ** | 0.00 NS | 0.79 ** |
| Replication | 4 | 16.25 NS | 2.58 NS | 0.22 NS | 0.04 NS | 9.39 NS | 0.002 NS | 0.01 NS | 0.004 NS | 0.006 NS |
| Seed coat color | 1 | 640.0 ** | 44.44 * | 0.08 NS | 0.02 NS | 650.28 ** | 0.05 ** | 0.306 ** | 0.004 NS | 0.72 ** |
| Replication | 4 | 3.6 NS | 7.2 NS | 0.12 NS | 0.04 NS | 5.24 NS | 0.002 NS | 0.008 NS | 0.002 NS | 0.007 NS |
| Sowing depths | 2 | 2211.47 ** | 78.92 ** | 0.49 ** | 0.02 ** | 116.49 * | 0.32 ** | 0.11 * | 0.009 * | 0.31 ** |
| Replication | 4 | 7.07 NS | 3.31 NS | 0.17 NS | 0.04 NS | 3.69 NS | 0.005 NS | 0.04 NS | 0.004 NS | 0.06 NS |
| Seed orientation | 2 | 1198.87 ** | 14.62 * | 0.06 NS | 0.02 NS | 41.22 NS | 0.03 ** | 0.25 ** | 0.02 * | 0.66 ** |
| Replication | 4 | 1.27 NS | 5.93 NS | 0.05 NS | 0.03 NS | 9.67 NS | 0.002 NS | 0.005 NS | 0.006 NS | 0.006 NS |

DF = degrees of freedom, * Significant at $p < 0.05$, ** Significant at $p < 0.01$, NS = Non-significant value.

**Table 7.** Two-way analysis of variance (ANOVA) for seed germination, growth attributes, and biomass allocation of *B. variegata* under different seed sizes, colors, sowing depths, and sowing orientation.

| Source of Variation | DF | Germination (%) | Shoot Length (cm) | Collar Diameter (mm) | Number of Leaves | Root Length (cm) | Shoot Dry Wt (g) | Root Dry Wt (g) | Leaves Dry Wt (g) | Total Dry Wt (g) |
|---|---|---|---|---|---|---|---|---|---|---|
| Seed size | 1 | 435.60 * | 891.70 ** | 2.89 ** | 170.40 ** | 695.89 ** | 0.10 ** | 0.02 ** | 0.03 ** | 0.45 ** |
| Replication | 4 | 51.40 NS | 1.04 NS | 0.14 NS | 1.09 NS | 2.17 NS | 0.001 NS | 0.00 NS | 0.001 NS | 0.002 NS |
| Seed coat color | 1 | 409.60 ** | 37.95 ** | 0.33 * | 2.03 NS | 196.43 ** | 0.007 ** | 0.01 ** | 0.004 * | 0.07 ** |
| Replication | 4 | 38.40 NS | 4.23 NS | 0.05 NS | 1.45 NS | 2.84 NS | 0.001 NS | 0.00 NS | 0.001 NS | 0.006 NS |
| Sowing depths | 2 | 2211.47 ** | 311.42 ** | 0.01 NS | 0.06 NS | 172.05 ** | 0.02 NS | 0.006 NS | 0.03 ** | 17.73 ** |
| Replication | 4 | 7.07 NS | 6.84 NS | 0.03 NS | 0.62 NS | 5.25 NS | 0.005 NS | 0.009 NS | 0.000 NS | 0.01 NS |
| Seed orientations | 2 | 1198.87 ** | 532.95 ** | 0.25 ** | 0.49 NS | 84.77 ** | 0.009 ** | 0.005 ** | 0.004 ** | 0.05 ** |
| Replication | 4 | 1.27 NS | 16.52 NS | 0.07 NS | 1.06 NS | 12.68 NS | 0.007 NS | 0.001 NS | 0.001 NS | 0.01 NS |

* Significant at $p < 0.05$, ** Significant at $p < 0.01$, NS = Non significant value.

## 4. Discussion

Seeds of *Bauhinia* species exhibited optimum germination at 25 °C and decreases at 20 °C and 15 °C temperature regimes. Low germination at a lower temperature is attributed to protein denaturation [23,24]. The natural occurrence of these tree species is extended from the tropical to the subtropical region in Garhwal Himalaya and the optimum temperature occurs during the rainy season in these areas [25] oscillating between 25 and

35 °C, which is the most suitable for natural regeneration. In both study species, the GI decreased, whereas MGT increased with increasing constant temperature regimes, as also recorded for *Albizia lebbek* (L.) Benth. of the family Fabaceae [26], and *Celtis australis* L. of the family *Cannabaceae* [27].

The quicker and more consistent germination in both *Bauhinia* species was achieved at 25 °Ctemperature regimes. Similar results were reported for *Dalbergia sissoo* Roxb. (Fabaceae) [28], and *Terminalia* spp. (Combretaceae) [29]. The rate of germination in most plant species seem to be strongly temperature dependent, but it is also related to other external factors: light, humidity, and seed coat width [12,30]. Zangoie et al. [31] conducted a study to examine the effects of temperature on seed germination of *Ferula assafoetida* L. (Umbelliferae) and revealed that the seed germination was optimum at 25 °C but not significantly higher as compared to the 20 °C temperature. A temperature range of 25 to 30 °C was required to obtain uniform and maximum germination in twelve indigenous and eight exotic multipurpose leguminous species [32]. The rate of germination and germination capacity increased with increasing temperature from 13 °C to 33 °C with the optimum at 28 °C [33]. Thus, studies on seed germination indicated that a certain range of temperature gives the highest percentage of germination; however, the optimum temperature requirement of each species is different for their germination, signifying a possible genetic control [34].

In the case of observations regarding seed storage under the present investigation, the seeds of the two *Bauhinia* species stored at room temperature, in different types of containers, resulted in reasonable viability and germination of seeds which indicates that storage at room temperature under climatic conditions of the present study site is effective for storage of *Bauhinia* seeds for one year. In an earlier study, Singh et al. [35] also observed that *Casuarina equisetifolia* L. (Casuarinaceae) seeds could be stored well under ambient conditions. Abdul-Banki and Anderson [36] reported that the low germination capacity and viability in seeds stored in cloth bags were caused due to changes in the physiochemical condition of seeds, chiefly the metabolism of seeds owing to the decrease in moisture content. The changes in seed metabolism are earlier reported as one of the major factors for low seed germination and viability [37]. We also observed that the germination and viability percentage of both *Bauhinia* species were recorded higher in polythene bags under room temperature and lower viability was found in cotton bags. The results of our study are at par with that reported for *Toona ciliata* M. Roem (Meliaceae) [38]. The seeds stored in cotton bags resulted in the loss of seed viability due to the loss of seed moisture during storage, which further affected the longevity of the seeds. A seed package that is moisture-proof or moisture-resistant would have extra importance in prolonging the viability and vigor of the seeds [23].

Seeds of *Bauhinia* that were sown with radicle downwards had higher germination percentages and produced larger seedlings than seeds that were sown with radicle facing upward and horizontal orientations. Contrary to the present study, seeds of *B. retusa* [15], *B. vahlli* Wight and Arnott, and *B. racemosa* Lamk. [39] had resulted in higher germination percentage, faster rates of germination, and seedling growth when sown with the radicle end facing upward direction in the soil surface. Similarly, the earliest onset of emergence in *Quercus leucotrichophora* A. Camus (Fagaceae) was recorded for seeds sown with radicle end-up [40]. In contrast, seeds of *Pinus roxburghii Sarg.* (Pinaceae) [41] and *Shorea robusta Gaertn* (Dipterocarpaceae) [42] had reported higher germination for seeds sown with radicle end downward. Swaminathan et al. Moreover, [43] recorded the maximum germination in *Derris indica* Lam. Bennet (Fabaceae) seeds when sown in a vertical position with the radicle end downwards compared to any other position. Singh et al. [18] reported that seed emergence of *Cinnamomum tamala* (Buch.-Ham.) T. Nees & Eberm. (Lauraceae) was higher in upright seed orientation as compared to the seeds sown in an inverted or horizontal orientation.

Krishna et al. [44] reported that purple color seeds have significant differences in seed germination and speed of germination over the other seed coat colors in *Erythrina indica Lam.* (Fabaceae). These findings are in agreement with the results of the present study

in which seed coat color had significantly influenced the germination, MGT, and GI, as well as growth attribute and biomass allocation in both *Bauhinia* species. Liu et al. [45] reported that the seeds having yellow seed coats of *Prunus armeniaca* L. (Rosaceae) had the highest germination percentage. Bonner et al. [46] also examined that seed coat color was pretentious for indicating the germination in *Gleditsia triacanthos* L. (Fabaceae) seeds. In the present study, the seeds of both *Bauhinia* species with a yellow seed coat color were found to be suitable for the production of quality seedlings in the nursery.

The size of an individual seed may be influenced by the physical condition of parent plants and affects the regeneration process of the population [47], seedling biomass [47], and seedling survival [48,49]. Germination of seeds is essential for the production of plants and to obtain the optimal number of seedlings for the regeneration of a species and afforestation/reforestation in their natural habitat [50]. Therefore, it is important to identify the consequences of seed size on germination and seedling growth of *Bauhinia* species for facilitating its large-scale plantations. The large-size seeds augment the emergence, and survival growth of the seedlings and biomass allocation as compared to the small-size seeds in both *Bauhinia* species. Based on the present study, large size seeds of both *Bauhinia* species supplemented the emergence, growth, and pertaining biomass of the seedlings compared to the small size seeds. Baraloto et al. [51] reported that the larger-seeded *Eperua grandiflora* (Aubl.) Bailland *Vouacapoua americana* Aubl. (both Caesalpiniaceae) produced larger seedlings as compared to small size seeds. Reich et al. [52] also reported that the seed mass significantly influences the emergence of seedlings and their survival in stressful environmental conditions and enhance flexibility in shoot/root allometry, speed of germination, and overall growth.

Temperature, moisture, oxygen, and soil conditions are essential for seed germination, emergence, and seedling recruitment [53,54] and these factors vary under different burial depths and influence the emergence and seeding establishment. Seeds sown at deep depth take more time for seedling emergence [55]. Ahirwar et al. [56] revealed the emergence of seedlings in *Butea frondosa* Roxb. Ex willd (Fabaceae) seeds were higher when sown at 2 cm depth and the emergence of seedlings was gradually reduced as the sowing depth was increased. Seed germination and other growth characteristics in *Vicia faba* L. (Fabaceae) decreased with increasing sowing depth [57]. The result of an earlier study showed that the upper soil layer had produced a higher seedling emergence rate, which indicates the favorable micro-habitat, light, water, nutrient, and pathogen for the germination of seeds of *Azadirachta indica* A. Juss. (Meliaceae) [17]. Seeds buried deeply resulted in delayed emergence of seedlings than those that were shallowly buried because the seeds buried in deep lost maximum reserves of nutrients in elongation of the shoot to come in the apices to reach and emerge in the soil surface as compared to a seed buried in shallower depths [58–60].

The seeds sown in an upright situation revealed detrimental effects on the emergence and seedling growth in our observations. The germination was postponed or suppressed by the upright positioning of seeds owing to the requirement of extra energy for the emergence of plumule from the soil when sown in this position. The epigeal type of seed emergence in *Hardwickia binata* Roxb. (Fabaceae) used additional energy for hypocotyl development [61]. Mahgoub [62] also reported similar results in *Delonix regia* (Bojer) Raf. and *Vachellia nilotica* (L.) P.J.H.Hurter & Mabb (both Fabaceae). The unusual morphological development of seedlings was the manifestation of the chemical substances or physiological changes due to the transposition of the seedling's strength [63].

The production of seedlings and nursery management are very important parts of most plantation programs which facilitate the landholders and farming communities with high-quality planting material [64,65]. *B. retusa* and *B. variegata* are ecologically and economically important species of Indian Himalaya, with large potential in afforestation or forest plantation programs. The improvement in the germination rate of forest species for production purposes is more crucial keeping allowance for the fact that forest tree seeds are less available commercially, are difficult to procure, and have low rates of germina-

tion [66–68]. Therefore, the initial step toward addressing these problems is to improve our knowledge and understanding of the major factors which affect seed germination and seedling growth [69]. However, the present study investigated the effects of temperature, seed size, and sowing position on germination and seedling growth of *B. retusa* and *B..* seeds, as well as the effects of storage containers on seed viability with the aim to provide assistance for quality seedling production. However, the changes in seed moisture during storage may also be addressed. The moisture content plays an important role in seed physiology in two ways. First, the ratio of vicinal water in the tissue of the seed to the bulk water may serve as the indicator of the expected metabolic activity of the seed which includes development, resting, and germination. Therefore, the absolute moisture content does not specifically dominate the physiological activity. Second, the extra dehydration may switch off the reserve material's production irreversibly and the senescence may be initiated that continues at varying pace till the availability of seed reserve [70]. Furthermore, apart from seed traits, the seedling emergence from the soil is also influenced by different environmental factors like light, temperature, pH, and other soil properties [71] and the initiation of seed germination generally needs specific soil conditions [72]. For example, seed germination occurs only in the condition when surface soil water conditions are optimum [73]. Thus, the monitoring of changes in seed moisture during storage and consideration of soil characteristics remains the limitation under the present investigation which may be the point of consideration for further studies while keeping the other factors like seed storage container, temperature, seed size, sowing position etc as constant.

## 5. Conclusions

The results of the present study indicate that the large-size seed had more germination percentage, as well as seedling growth and biomass production in both *Bauhinia* species. Yellow seed coat color is the best indicator for the maximum germination and seedling growth and biomass in both *Bauhinia* species. Seedling emergence of *B. retusa* and *B. variegata* was higher when sown at 4 cm and 2 cm soil depth, respectively. While their seedling growth characters and biomass attributes were the highest at 2 cm depth. Similarly, seed sowing orientation showed the highest emergence of seedlings at the upright position in *B. retusa* and the horizontal position in *B. variegata* with their growth and biomass. Thus, it may be rational to take into consideration, the size, appearance, sowing depth, and direction of sowing of the seeds in *B. retusa* and *B. variegata* for the production of good quality seedlings to continue metabolic integrity and sub-cellular management even under an unfavorable situation.

**Author Contributions:** N.Y. experimented, data collection, and prepared the first draft of the manuscript. B.S. and C.S.D. analysed data and preparation of the first manuscript. V.P.K. and B.S. designed the research and supervised the research. B.S., D.R., V.P.K. and M.K. reviewed and modified the manuscript. M.K.R. and T.A. used software to analyse the data. All authors have read and agreed to the published version of the manuscript.

**Funding:** This research received no external funding.

**Data Availability Statement:** The data obtained during the investigation in this study are included in the article.

**Acknowledgments:** The authors are thankful to the College of Forestry, V.C.S.G Uttarakhand University of Horticulture and Forestry, Ranichauri, Tehri Garhwal, Uttarakhand for providing all the facilities.

**Conflicts of Interest:** The authors declared as an apparent no conflict of interest.

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
