# Peer review of "Effect of Temperature, Seed Size, Sowing Depth, and Position on Seed Germination and Seedling Growth of Bauhinia retusa Roxb. and Bauhinia variegata L."

_forests, doi:10.3390/f14081664_

Round 1

Reviewer 1 Report

Bauhinia retusa and Bauhinia variegata are ecologically and economically important species, with large potential in afforestation or plantation programs. This study investigated effects of temperature, seed size, and sowing position on germination and seedling growth of Bauhinia retusa Roxb. and Bauhinia variegata L. seeds, as well as effects of storage containers on seed viability, aimed to provide help for quality seedling production. Although the authors did their best to design experiments, but necessary condition control was absent, resulting in low scientific value of this work.

Roughly, this study was composed of two parts. The one is seed storage. For seed storage, seed moisture and storage temperature are two key factors to affect seed longevity. But in this study, seed initial moisture contents were not determined, and changes in seed moisture during storage were not monitored. Furthermore, the seeds were stored openly, consequently we do not know whether these lost moisture (in drier air) or gain moisture (in wetter air). Thus, we can not judge the response of seed viability to seed moisture change and seed storage behaviour category.

The other is seedling emergent in nursery. Similarly, necessary condition control was absent in this study. Seedling emergent from soil is a result of the interaction between seeds and soil conditions, which is determined by seed traits, including seed size, germination requirement for light, temperature and water, on the hand, and by soil condition, such as soil traits, soil moisture, temperature, and their changes in different depth. But the authors determined none of them.

As some necessary condition control was absent, this work is unrepeatable, I think. 

Author Response

Bauhinia retusa and Bauhinia variegata are ecologically and economically important species, with large potential in afforestation or plantation programs. This study investigated the effects of temperature, seed size, and sowing position on germination and seedling growth of Bauhinia retusa Roxb. and Bauhinia variegata L. seeds, as well as the effects of storage containers on seed viability, aimed to provide help for quality seedling production. Although the authors did their best to design experiments, necessary condition control was absent, resulting in the low scientific value of this work.

Sorry for the inconvenience the manuscript has now been improved substantially. 

Roughly, this study was composed of two parts. The one is seed storage. For seed storage, seed moisture, and storage temperature are two key factors to affect seed longevity. But in this study, seed initial moisture contents were not determined, and changes in seed moisture during storage were not monitored. Furthermore, the seeds were stored openly, consequently, we do not know whether these lost moisture (in drier air) or gain moisture (in wetter air). Thus, we can not judge the response of seed viability to seed moisture change and seed storage behavior category.

Thank you for your valuable suggestion. Now the manuscript improved substantially and the moisture content is not included at this time but surely be included in the other study in the Himalayan region.

The other is seedlings emergent in the nursery. Similarly, necessary condition control was absent in this study. Seedling emergence from the soil is a result of the interaction between seeds and soil conditions, which is determined by seed traits, including seed size, germination requirement for light, temperature, and water, on the hand, and by soil condition, such as soil traits, soil moisture, temperature, and their changes in different depth. But the authors determined none of them.

Thank you for your valuable suggestion. Now the manuscript improved substantially, the control condition for the seedling emergence not considered in this study but your valuable comments are should help us to conduct future studies on the same line for the valuable species of the Himalayan region.

As some necessary condition control was absent, this work is unrepeatable, I think. 

Sorry for the inconvenience the manuscript has now as been improved substantially and now it is suitable for publication in the journal. 

Reviewer 2 Report

Manuscript Forests-2459110: Temperature, seed size, and sowing position influenced the seed germination and seedling growth of Bauhinia retusa Roxb. and Bauhinia variegata L.

This study and its presentation provide great value in our understanding of Bauhinia Spp requirements for germination in restoration projects. This work provides tools to produce quality planting material for Bauhinia Spp in the nursery for restoration programs. However, the manuscript needs more work and I consider that the manuscript is ready for publication after the following comments:

Minor revisions:

L23. What were the temperatures to which they were exposed?

L101. Following objectives…

L111. What temperature was experienced by the seeds at room temperature?

L115. replace at for the abbreviation i.e. In the same line, correct the symbol of degree (ºC)

L129. Check the degree (ºC) symbol in all manuscript.

L136. Define FYM or what it means.

The statistical approach used to analyze the results is not mentioned in the materials and methods. Add section and describe how the results will be analyzed.

 L193. Specify what is on the y-axis and its units. Also, add the standard deviations of the columns and in the figure legends. The same for all figures.

L194. In all figures, at the end of the sentence, once the SD is added: The vertical bars indicate the standard deviation of the mean (±SD).

Author Response

L23. What were the temperatures to which they were exposed?

Ans: The temperatures to which the seeds were exposed are now mentioned in L23.L101. Following objectives…

Ans: Thank you for the suggestion, necessary correction has been incorporated following the reviewers suggestion.

L111. What temperature was experienced by the seeds at room temperature?

Ans: Thank you for the valuable suggestion. The room temperature experienced during the study period has now been incorporated in the line mentioned by the reviewer.

L115. Replace at for the abbreviation i.e. in the same line, correct the symbol of degree (ºC)

Ans: Thank you for the valuable suggestion. Necessary corrections about changing the abbreviation i.e. “at” and correcting the symbol ºC have been made as indicated by the reviewer.

L129. Check the degree (ºC) symbol in all manuscripts.

Ans: The symbol has now been corrected throughout the manuscript. We are thankful for the valuable input of the reviewer. 

L136. Define FYM or what it means.

Ans: Thank you for the suggestion. The full form of FYM has now been incorporated into the manuscript. The authors feel this full form is self-explanatory to understand the term for the scientific community/readers and defining it is not very necessary. 

The statistical approach used to analyze the results is not mentioned in the materials and methods. Add a section and describe how the results will be analyzed.

Ans: Thank you for your suggestion. The statistical analysis has been now incorporated into the material and methods section.

 L193. Specify what is on the y-axis and its units. Also, add the standard deviations of the columns and in the figure legends. The same for all figures.

Ans: Thank you for your valuable suggestion. Now all the necessary medication has been incorporated.

L194. In all figures, at the end of the sentence, once the SD is added: The vertical bars indicate the standard deviation of the mean (±SD).

Ans: The entire figure has been now modified as per your suggestion.

Reviewer 3 Report

The work entitled “Temperature, seed size, and sowing position influenced the seed germination and seedling growth of Bauhinia retusa Roxb. and Bauhinia variegata L.” presents good results, but some points should be reviewed, as exemplified below:

The title of the manuscript is coherent.

The abstract must be rewritten after carrying out the recommended revisions.

Introduction

In the first paragraph, it is necessary to add quotes in several sentences. For example, it is described: “Bauhinia (family Fabaceae) is a large genus of flowering plants that naturally occur in subtropical forest ecosystems with various forms such as trees, shrubs, and climbers.” What is the quote for this sentence? Include citations and references throughout the manuscript where necessary. Follow this example.

The authors need to make clear the problem of the study. The introduction can be structured as follows:

_ Importance of species (the first paragraph only needs to be revised regarding citations);

_ What was the problem that led to the study;

_ The importance of advancing knowledge for these aspects studied (temperature, size, and sowing position);

_ Purpose of the study (it is currently well described).

Methodology

Describe what mature pods are... For this, did you follow the development or did you consider the color change? How do you know they were ripe pods?

Line 115 reads “250C”, correct for the correct unit of measure; this must be in degrees. Also, it must be overwritten! Check these inconsistencies throughout the manuscript and correct them.

What is the humidity of the Seed Germinator Machine? What are the characteristics of this Seed Germinator used in the experiment? Is there humidity control?

Change in all text "percent" to "percentage".

Line 148: “the stranded procedures.” What does it mean?

It would be interesting if the authors could add the morphology of seeds and seedlings of both species.

At the end of the methodology, the authors must include the design of the experiments, experimental unit, and repetitions. In addition, they must report the statistical program used and which test was applied after the analysis of variance.

Results

In table 1-5. Add the stat test letters. Without the letters, it is not possible to decide which treatment is best.

In all tables separate the standard deviation by a space. This must be done before the ± symbol. Also, it needs to be described whether it is the standard deviation or standard error of the mean.

In all tables and figures captions make it clear which are the two species used in the study. Captions should be accurate and self-explanatory.

Tables 6 and 7 are unnecessary. Can be removed from the manuscript.

Figures 1-3 are not plotted properly. For example, deleting the chart title. This is unusual. Add labels on the abscissa and ordinate axis.

After corrections, the text must be revised by a specialist in the language.

Author Response

  1. The title of the manuscript is coherent. The abstract must be rewritten after carrying out the recommended revisions.

Ans: Thank you for the valuable suggestion. We have incorporated the necessary corrections and changes in the abstract as suggested by the reviewer.

  1. In the first paragraph, it is necessary to add quotes in several sentences. For example, it is described: “Bauhinia (family Fabaceae) is a large genus of flowering plants that naturally occur in subtropical forest ecosystems with various forms such as trees, shrubs, and climbers.” What is the quote for this sentence?

Ans: Thank you for the valuable suggestion. Following the recommendation of the reviewer we have now added the necessary references in the introduction section wherever it was possible.

  1. Include citations and references throughout the manuscript where necessary.

Ans: We appreciate the critical suggestion of the reviewer. As per the directions of the reviewer, we have now added the required references in the manuscript throughout.

  1. The authors need to make clear the problem of the study. The introduction can be structured as follows:

 Importance of species (the first paragraph only needs to be revised regarding citations);

what was the problem that led to the study;

The importance of advancing knowledge for these aspects studied (temperature, size, and sowing position);

Purpose of the study (it is currently well described).

Ans: Thank you for the worthy suggestion. Necessary changes in the introduction section have been made as per the recommendations of the reviewer.

  1. Methodology: Describe what mature pods are... For this, did you follow the development or did you consider the color change? How do you know they were ripe pods?

Ans: Thank you for the suggestion. The pods which were hard, flat, and brown in appearance were considered mature pods. The same has been indicated in the manuscript for better understanding to the readers.

  1. Line 115 reads “250C”, correct for the correct unit of measure; this must be in degrees. Also, it must be overwritten! Check these inconsistencies throughout the manuscript and correct them.

Ans: We appreciate the valuable suggestion. Necessary corrections have been incorporated in the manuscript.

  1. What is the humidity of the Seed Germinator Machine? What are the characteristics of this Seed Germinator used in the experiment? Is there humidity control?

Ans: We appreciate the critical suggestion of the reviewer. As per the directions of the reviewer, we have now added the humidity percentage in the manuscript.

  1. Change in all text "percent" to "percentage".

Ans: Thank you for your worthful suggestions. The percent has been now changed as a percentage throughout the manuscript.

  1. Line 148: “The stranded procedures.” What does it mean?

Ans: There was a typographic error in the statement. “the stranded procedures” has been corrected to “The standard procedures”.

  1. It would be interesting if the authors could add the morphology of seeds and seedlings of both species.

Ans: Thank you for your worthful suggestions. The morphology of the seed and seedling of both species has been now incorporated in the manuscript.

  1. At the end of the methodology, the authors must include the design of the experiments, experimental units, and repetitions. In addition, they must report the statistical program used and which test was applied after the analysis of variance.

Ans: Thank you for your suggestion. The statistical analysis has been now incorporated into the material and methods section.

  1. In table 1-5. Add the stat test letters. Without the letters, it is not possible to decide which treatment is best.

Ans: Thank you for your suggestion. The statistical test letter has been now incorporated into all the tables.

  1. In all tables separate the standard deviation by a space. This must be done before the ± symbol. Also, it needs to be described whether it is the standard deviation or standard error of the mean.

Ans: Thank you for your valuable suggestion. Now the standard derivation is separated by the space and described in the figure and table.

  1. In all tables and figures captions make it clear which are the two species used in the study. Captions should be accurate and self-explanatory.

Ans: We appreciate the valuable suggestion. Necessary corrections have been incorporated in the caption of the table and figure.

  1. Tables 6 and 7 are unnecessary. Can be removed from the manuscript.

Ans: We appreciate the valuable suggestion.  But the other two reviewers did not give suggestions to remove these tables, So, we not removed the tables.

  1. Figures 1-3 are not plotted properly. For example, deleting the chart title. This is unusual. Add labels on the abscissa and ordinate axis.

Ans: Now we have improved Figures 1-3 as per your suggestion.

  1. Comments on the Quality of English Language. After corrections, the text must be revised by a specialist in the language.

Ans: Thank you for your worthy recommendation. Now we have improved the statements of the manuscript to the best of our ability and knowledge of the English language.

Reviewer 4 Report

The article is scientifically interesting but needs to be improved:

General comments: scientific names are written in Italics, and include the Authority and the plant family; words and bibliographic references are in Bold along the text.

Introduction, bibliographic references are needed along the text. The article has not an objective but two questions to be reply.

Materials and Methods, include MGT and GI formulas, and the statistical analysis is not developped at all.

Results, Tables and Figures ought to be improved with a title and an explanatory legen; they shoul be clear enough to the reader, without having to read the whole article. Tables need to explain all the abbreviations, symbols, etc.; some senteces and paragraphs are hard to understan, hence, syntax needs to be improved.

Discussion, some parts are confusing, need to improve syntax and the whole organization of the ideas. All the statistical analysis information is found here and not in the Material and Methods section, thus, this section needs to be restructured. Through the text, there are many assumptions (or hypothesis or speculations) presented without any evidence to support them or, even, affirmative sentences without context or a conceptual frame, that is why there are many "And???" in this section. The last paragraph is a conclusion, therefor, authors can developped a small Conclusion with those ideas.  

References, they are congruent with the aim of the article.

Please, see file attached.

Author Response

  1. The article is scientifically interesting but needs to be improved:

General comments: scientific names are written in Italics, and include the Authority and the plant family; words and bibliographic references are in Bold along the text.

Ans: Thank you for the valuable suggestion. The scientific names now have been done in italics and authority, plant family has been now incorporated; words and bibliographic references have been corrected as per the Journal style.

  1. Introduction, bibliographic references are needed along the text. The article has not an objective but two questions to reply to.

Ans: We appreciate the worthful suggestions of the reviewer. As per the recommendation, we have now added the bibliographic references in the introduction section where ever applicable. Also, the objectives have been modified.

  1. Materials and Methods include MGT and GI formulas, and the statistical analysis is not developed at all.

Ans: Thank you for the valuable suggestion. The description of the statistical analysis used in the study has now been added in the material and methods section.

  1. Results, Tables, and Figures ought to be improved with a title and an explanatory legend; they should be clear enough to the reader, without having to read the whole article.

Ans: We appreciate the worthful suggestions of the reviewer. Now the results, table, and figure have been improved.

  1. Tables need to explain all the abbreviations, symbols, etc.; some sentences and paragraphs are hard to understand, hence, syntax needs to be improved.

Ans: Thank you for the valuable suggestion. All the necessary improvement has been done to the tables and manuscripts as per your suggestions.

  1. Discussion, some parts are confusing and need to improve the syntax and the whole organization of the ideas. All the statistical analysis information is found here and not in the Material and Methods section, thus, this section needs to be restructured. Throughout the text, there are many assumptions (or hypotheses or speculations) presented without any evidence to support them or, even, affirmative sentences without context or a conceptual frame, which is why there are many "And???" in this section.

Ans: Thank you for the valuable suggestion. Following the advice of the reviewer, we have restructured and modified the discussion part to improve the quality of the section and for a better understanding of the readers.  

  1. The last paragraph is a conclusion; therefore, authors can develop a small Conclusion with those ideas.  

Ans: Thank you for the worthy suggestion. The last paragraph has now been reframed to provide a conclusion of the study.

  1. References are congruent with the aim of the article.

Ans: The authors extend thanks to the reviewer for the remark.

Thank you for improving the quality of the manuscript. All the corrections and suggestions given by the reviewers in the PDF file of the manuscript also now has been included in the revised manuscript.

Round 2

Reviewer 1 Report

I checked the revised MS and authors' response to my comments, I still think that the experiments were not properly controlled, such determination and control of seed moisture.Thus it is difficult to evaluate its scientific values. Although the authors respond to my comments, there are few substant improvement compared with its original version. My suggestion is Rejection.

Author Response

Comments: I checked the revised MS and authors' response to my comments; I still think that the experiments were not properly controlled, such as the determination and control of seed moisture. Thus, it is difficult to evaluate its scientific values. Although the authors respond to my comments, there are few substant improvements compared to its original version. My suggestion is Rejection

Answer: Authors are thankful for your valuable suggestions to improve the overall quality of the manuscript. However, the suggestions of the reviewer could not be incorporated into the manuscript because the experiment was not conducted during the study. Our submission in the manuscript is that control of seed moisture for the storage of seeds could be important. Although, it is not generally preferred by the farmer to adopt such conditions. Therefore, it was not the focus of the study. The study was conducted to examine how quality seedlings could be produced at the minimum cost. Thus we could not be considered your valuable suggestions in the manuscript in its present form, but in future studies, we definitely incorporate your valuable suggestions in our studies. We hope you will consider the present data of the manuscript for the final publication of the manuscript.

Reviewer 4 Report

Authors have done all the corrections suggested; however, there are some details, through the text, that need to be corrected, mainly in the Reference section.

Please, see the file attached.

Author Response

Comments: Authors have done all the corrections suggested; however, there are some details, in the text, that need to be corrected, mainly in the reference section.

Answer: Thank you for your positive input. Your valuable suggestions have been incorporated in all the corrections for the improvement of the overall quality of the manuscript as suggested.